# Resilience across time: Longitudinal measurement invariance of the Brief Resilience Scale in French university students

Anja Todorović[1,2]*, Cédric Baumann[1,2], Christine Rotonda[1,3], Elise Eby[3], Sarah Lapcevic[1], Stéphanie Bourion-Bédès[4,5]

1 UMR U1319 INSPIIRE, University of Lorraine, Inserm, Nancy, France, 2 Methodology, Data Management and Statistics Unit, University Hospital of Nancy, Nancy, France, 3 Centre Pierre Janet, University of Lorraine, Metz, France, 4 Team DevPsy, Inserm U1018, CESP, University Paris-Saclay, UVSQ, Villejuif, France, 5 University Department of Child and Adolescent Psychiatry, Versailles Hospital, Versailles-Le-Chesnay, France

* anja.todorovic@univ-lorraine.fr

## Abstract

### Purpose

The response shift effect occurs in Patient-reported outcome measures (PROMs) as a consequence of internal changes in perception of the measured concept. The purpose of this study was to assess the potential presence of the response shift effect within the French version of the Brief Resilience Scale (BRS) in a population of university students in the Lorraine region of France.

### Methods

To measure resilience, the French version of the BRS was used. To examine the potential presence of the response shift effect within this questionnaire, the Oort's procedure and Rasch Measurement Theory based ROSALI-RMT algorithm was used.

### Results

The BRS demonstrated longitudinal measurement invariance across Time 0 and Time 1, as assessed using the ROSALI algorithm. A slight but significant increase in resilience was observed over time (time effect = 0.22, p < 0.0001), with no evidence of response shift effects, suggesting that the observed changes in resilience were genuine and not influenced by recalibration.

### Conclusion

These findings support the validity of the BRS in measuring resilience over time and highlight its utility in capturing genuine changes in resilience.

**Data availability statement:** The dataset contains human research participant data collected from French university students as part of a longitudinal psychological study. Although the data have been coded and direct identifiers removed, the combination of variables (including the longitudinal design, demographic characteristics, and repeated psychological measures) means that full anonymization cannot be guaranteed. Participants did not provide explicit consent for public data sharing at the time of data collection. In accordance with European data protection regulations (GDPR) and institutional data protection requirements, unrestricted public dissemination of the dataset is therefore not permitted. For these ethical and legal reasons, the dataset cannot be made publicly available. However, the data are available upon reasonable request for researchers who meet the criteria for access to confidential data and agree to comply with applicable data protection requirements. Data access requests may be directed to: Professor Nelly Agrinier Research Unit Director UMR1319 INSPIIRE Université de Lorraine nelly.agrinier@univ-lorraine.fr The minimal dataset necessary to reproduce the findings reported in the manuscript will be shared with approved researchers under appropriate data protection conditions.

**Funding:** This research is a part of a PhD project funded by the French Network of Doctoral Programmes in Public Health (RDSP), coordinated by EHESP French School of Public Health. The funders had no role in study design, data collection and analysis, decision to publish, or preparation of the manuscript.

**Competing interests:** The authors have declared that no competing interests exist.

## Introduction

Psychological resilience is commonly defined as the capacity to adapt positively to adversity and to maintain or regain psychological functioning following stressful or traumatic experiences [1,2]. Importantly, resilience does not imply the absence of distress or the uninterrupted preservation of well-being during adversity, but rather reflects an individual's ability to recover, adjust, and continue functioning despite temporary disruptions. Resilient individuals typically exhibit strong emotional regulation, effective coping strategies, and healthy social connections, all of which help them navigate difficult situations more effectively [3].

Traditionally viewed as a stable personal trait, resilience is now increasingly understood as a dynamic and context-dependent process that can evolve over time. It is not a fixed characteristic, but rather something that can be cultivated through strategies like emotional regulation, cognitive reprogramming, and social support [2,4]. This shift in perspective underscores the importance of recognizing resilience as a modifiable resource that can be developed and strengthened, rather than simply an inherent quality.

The COVID-19 pandemic dramatically tested individuals' resilience on a global scale. As the virus spread and lockdowns were enforced, people were forced to adapt to new, often challenging circumstances. Health risks, economic uncertainty, and social isolation created immense psychological pressure [5]. University students, who are often in a period of transition and identity formation [6], faced disruptions in their academic lives, changes in living arrangements, and the abrupt shift to remote learning. For many students, this resulted in heightened stress related to uncertainties about education, career prospects, and social connections [7]. The lack of social support, common in university settings, contributed to feelings of isolation and anxiety [8].

During this time, resilience became a critical psychological resource. Multiple studies have shown that individuals with higher levels of resilience were better equipped to cope with pandemic-related stressors and consistently reported lower levels of anxiety, depression, and psychological distress during the COVID-19 pandemic [9–12]. This underscores the importance of resilience in helping individuals maintain stability during times of crisis, as resilient individuals tend to mobilize adaptive coping strategies, regulate emotions more effectively, and draw on social support to manage stress and uncertainty [2,13]. However, to accurately assess resilience, it is crucial to consider how it is measured.

Historically, resilience was viewed as a stable personal trait linked to an individual's innate ability to manage adversity [14,15]. However, recent research has shifted toward viewing resilience as a dynamic and evolving process. This has led to the development of various self-report instruments designed to assess resilience. These instruments reflect different theoretical perspectives, ranging from relatively stable trait-based conceptualizations to more dynamic, process-oriented approaches. Commonly used measures include the Connor-Davidson Resilience Scale (CD-RISC) [16], the Resilience Scale (RS) [17], and the Brief Resilience Scale (BRS) [18], which

differ in their emphasis on stable personal characteristics, protective resources, or recovery processes. In the present study, we focused on the BRS because it operationalizes resilience specifically as the capacity to bounce back from stress, a definition consistent with contemporary views of resilience as a dynamic and context-dependent process. Its brevity, unidimensional structure, sensitivity to change, and availability of a validated French version make it particularly suitable for longitudinal designs and repeated measurements in student populations during an evolving crisis such as the COVID-19 pandemic [18–20]. It is essential to assess whether these tools maintain consistent validity over time, especially during periods of significant stress like the COVID-19 pandemic.

While some resilience instruments have demonstrated cross-sectional measurement invariance groups (e.g., gender, age, education) [20–23], relatively few studies have examined longitudinal measurement invariance of resilience scales. To our knowledge, no prior study has specifically investigated how response shift might affect the longitudinal validity of resilience measures during prolonged stressors such as the COVID-19 pandemic. Response shift refers to changes in individuals' internal standards, values, or conceptualization of a construct over time, occurring through processes of recalibration, reprioritization, or reconceptualization, and may lead individuals to evaluate their resilience differently even if their underlying capacity remains unchanged [24]. Failure to account for such effects may result in misinterpretation of observed changes in resilience over time.

Despite these concerns, the BRS represents a theoretically and psychometrically robust candidate for longitudinal assessment, given its unidimensional structure, its focus on a general recovery capacity rather than context-specific appraisals, and its demonstrated internal consistency and stability across populations [18–20]. We therefore hypothesized that the BRS, when used across multiple time points, would maintain its psychometric properties, enabling valid comparisons of resilience over time. This study aimed to fill this gap by examining the longitudinal measurement invariance of the French version of the BRS within a population of university students in the Lorraine region during the COVID-19 pandemic.

## Methods

### Study design

This study is a psychometric analysis of the response shift effect of the data from the PIMS2-SPE study ("Feelings and psychological impact of the COVID-19 epidemic on students in the Grand Est area (France)"), an observational longitudinal study conducted via an online survey from 14 April 2021 to 31 January 2023, during the COVID-19 pandemic in France.

Given the approximate student population of the University of Lorraine of around 50,000 individuals, a random draw, stratified by gender and level of study, of 10,000 students was held 5 times, every two months from April to December 2021. Thus, these 5 waves of inclusions created the 5 study sub-samples. The follow-up of these sub-samples was the following: T0 – inclusion, T1 – at 6 weeks post-inclusion, T2 – at 12 weeks post-inclusion, T3 – at 18 weeks post inclusion, T4 – at 1 year post-inclusion. The measurement times used for this study were T0 and T1. Subsequent follow-up measures were not included in the present analysis because participant attrition became substantial from T2 onward, resulting in insufficient longitudinal data to reliably conduct measurement invariance and response shift analyses.

Detailed information regarding the purpose of the study was sent to all students, as well as a form for providing informed consent to participate in the study. The survey was completed anonymously to ensure the confidentiality and reliability of the data. All procedures were conducted following the principles of the Declaration of Helsinki. When we launched the study, our target was the entire student population in Lorraine, i.e., more than 50,000 students. It was estimated that 5–10% of the students would respond to the survey, i.e., somewhere between 2,500 and 5,000 students, which should have ensured ample statistical power.

This study was registered in the data processing register of the University of Lorraine with the Legal Affairs Department under the reference number 2021/150, dated March 16, 2021, in compliance with the General Data Protection Regulation (GDPR).

## Data collection

Data was gathered through an online survey [25]. Our team reached out to the deans' offices at the University of Lorraine, who then distributed the survey link to the entire student body via their university email addresses. The survey included questions on students' sociodemographic characteristics, their living and learning conditions during the lockdown, the perceived health threat of COVID-19 in their immediate environment, the impact of lockdown and related measures on their living and learning conditions, as well as self-administered questionnaires assessing their health status. The full survey questionnaire is available from the corresponding author upon reasonable request.

## Measures

**Sociodemographic and other data.** Data was collected pertaining to participants' sex, age, living conditions, university curriculum, as well as any specificities linked to the COVID-19 pandemic.

**Resilience.** The Brief Resilience Scale (BRS) is a 6-item, 5-point Likert scale [18] measuring levels of resilience, which is defined as the ability to bounce back or recover from stress. The participants are asked to state their level of agreement or disagreement with 6 statements, by choosing one of the following response modalities: "strongly disagree", "disagree", "neutral", "agree", "strongly agree". The response modalities are coded from 1 to 5 respectively for items 1, 3, and 5, and they are coded inversely from 5 to 1 for negatively worded items 2, 4, and 6. The total score is calculated by averaging the responses across all items. According to the framework proposed by Smith et al. [26], scores below 3 indicate low resilience, scores between 3 and 4.3 denote moderate resilience, and scores above 4.3 reflect high resilience [26]. The original version of the BRS exhibited good internal consistency (Cronbach's α = 0.80–0.91) and satisfactory test-retest reliability, with intra-class correlation (ICC) values of 0.62 and 0.69 [18].

The BRS was translated into French using a forward-backward translation process with cultural adaptation. A native French-speaking mental health expert carried out the forward translation, and an independent English-speaking translator performed the back translation. Any discrepancies were resolved by a bilingual author, and the translation was piloted with three midwives, requiring no further adjustments. The French version of the BRS (BRS-F) was validated in a midwifery sample, confirming its unifactorial structure and demonstrating strong psychometric properties [19].

A study conducted on a different sample of the same population as this study confirmed that BRS-F had a sound unifactorial structure and presented overall adequate goodness-of-fit indices. It showed a good level of internal consistency at 0.88 for Person Separation Reliability (PSR) and 0.86 for Cronbach's α. A further examination revealed no significant Differential Item Functioning (DIF) of the BRS-F in this population [20].

## Descriptive analysis

Quantitative variables were described by means and their standard deviation (SD), while qualitative variables were described by percentages.

## Confirmatory Factor Analysis (CFA)

The BRS-F questionnaire structure was examined through a CFA conducted using covariance-based structural equation modeling with maximum likelihood estimation and the Satorra-Bentler adjustment. The "lavaan" package in R was used [27]. The CFA aimed to assess the fit of the predefined unidimensional factor structure model. Model fit was evaluated using several indices, including the root mean square error of approximation (RMSEA), standardized root mean square

residual (SRMR), and comparative fit index (CFI). RMSEA, a parsimony-adjusted index, indicates a good fit when values are closer to 0. The SRMR measures the square root of the difference between the residuals of the sample covariance matrix and the proposed model. CFI compares the fit of the target model to an independent model, with values closer to 1 indicating a better fit. Models were deemed to have a good fit if RMSEA < 0.08, SRMR ≤ 0.07, and CFI > 0.9 [28]. Internal consistency reliability was assessed using Cronbach's alpha coefficient (α), where values between 0.8 and 0.89 were considered good, and values greater than 0.9 were considered excellent [29].

## Response shift analysis

The response shift analysis was conducted using ROSALI-RMT – a Rasch Measurement Theory based algorithm [30] which follows Oort's original 4-step procedure [31] for testing invariance between two times of measurement.

**Partial Credit Model (PCM).** A prerequisite to testing for response shift with the ROSALI-RMT algorithm is to verify that the data at the first time of measurement fits the Partial Credit Model. This analysis was conducted using the PCM module available at the PRO-online website [32,33].

**ROSALI-RMT.** The ROSALI-RMT module available at PRO-online was used to detect the presence of response shift [30,33]. This algorithm was adapted from the ROSALI-IRT algorithm using longitudinal partial credit models (PCM) to detect response shift based on Rasch Measurement Theory (RMT) [34]. It is based on Oort's procedure [31], which aims to test the global occurrence of response shift using a likelihood ratio test, and it observes the following 4 steps:

*Step 1*

The first step of Oort's procedure implies establishing a measurement model (Model 1) without any imposed constraints on response shift parameters across time.

*Step 2*

The second step constrains all response shift parameters to be equal across time, resulting in a model which assumes an absence of response shift (Model 2). Assuming longitudinal measurement invariance, this model is tested against the first model and compared using a Likelihood ratio test (LRT). The outcome of the LRT conditions the next step of the procedure. In the case of a significant results of the LRT, a global occurrence of response shift is assumed and the procedure continues on to step 3. If the results of the LRT are not significant, the procedure skips step 3 and goes directly on to step 4.

*Step 3*

The third step occurs if the LRT is significant, and it aims to determine the types of response shift present in the concerned dimensions. In this step, the model M is improved step-by-step by relaxing response shift parameter constraints one by one, resulting in Model 3 which includes all instances of detected response shift. In this step, recalibration detection is possible. To detect recalibration, a Bonferroni correction is applied to compare the results of the Wald tests of different models. To determine if recalibration is uniform or not, a Wald test is performed at 5% significance level.

*Step 4*

The fourth and final step of the procedure provides us with the Model 4, which corresponds to the third model if the LRT results were significant, or to the second model if the results were not significant. This fourth model estimates differences in latent trait means across time, which are potentially adjusted for identified response shift if applicable, to evaluate longitudinal change.

***Missing data handling***: The ROSALI-RMT model handles missing data by leveraging the specific objectivity of Rasch-family IRT models, which enables unbiased estimation of latent traits even when item responses are missing [35]. This property ensures parameter estimates remain valid regardless of missing data patterns [35–37]. Unlike Structural

Equation Modeling (SEM), which relies on assumptions such as missing completely at random (MCAR) or missing at random (MAR) and often employs imputation methods [38,39], ROSALI-RMT avoids imputation and uses all available data directly [35]. This makes ROSALI-RMT especially robust in the presence of potentially non-ignorable missing data (MNAR), as supported by findings from De Bock et al. [40,41] and Hardouin et al. [42].

## Ethics approval

Detailed information regarding the purpose of the study was sent to all students, as well as a form for providing informed written electronic consent to participate in the study. The survey was completed anonymously to ensure the confidentiality and reliability of the data. This study was performed in line with the principles of the Declaration of Helsinki. This study was registered in the data processing register of the University of Lorraine with the Legal Affairs Department under the reference number 2021/150, dated March 16, 2021, in compliance with the General Data Protection Regulation (GDPR).

## Results

### Sample description

The sociodemographic and academic characteristics, as well as living conditions, are reported in the Table 1. The sample consisted of 920 university students. All variables had less than 10% missing data, indicating an acceptable level of completeness. The mean age of the participants was 21.5 years (SD = 3.9). Regarding gender, 30.4% of the participants were male, while 68.7% were female. In terms of financial aid, 55.9% of the students were not receiving financial aid, and 43.5% were on a scholarship. When asked about part-time employment, 72.9% reported that it was not applicable to them, 7.0% had their activities interrupted during the lockdown, 7.1% reported an increase in activity, and 12.4% indicated no change in their work activities during the lockdown.

Regarding academic characteristics, the mean number of years since starting university was 3.0 years (SD = 3.4). The majority of students were enrolled in fields such as sports, medical sciences, and science and technology (43.6%), followed by social and human sciences (19.1%), law, economics, and management (13.7%), arts, humanities, and languages (12.4%), and other programs (11.1%). Additionally, 56.1% of the students were pursuing an undergraduate degree, 29.6% were graduate students, and 14.1% were in other degree programs.

In terms of living arrangements, 40.4% of the students lived with their parents, 34.0% lived alone, 12.8% lived with a partner, 10.9% lived in flatsharing situations, and 1.8% reported other living arrangements. For COVID-19 exposure within their households, 69.6% reported no one at home having had COVID-19, while 1.5% had confirmed and hospitalized cases, 17.8% had confirmed and non-hospitalized cases, and 10.9% had suspected cases. Regarding the students' relatives and acquaintances, 26.1% reported that no one in their network had COVID-19, while 14.2% reported confirmed and hospitalized cases, 52.7% reported confirmed and non-hospitalized cases, and 6.8% reported suspected cases.

### Resilience levels

The full sample consisted of 920 participants (Table 2). A total of 15.5% of data were missing at the second measurement time point. While this exceeds the commonly negligible threshold (<5%), it remains within the acceptable range for longitudinal designs. Given the robustness of the ROSALI-RMT algorithm and Rasch-family models to missing data, particularly under MAR or MNAR conditions, all available data were retained and analyzed without imputation. No statistically significant differences were observed between the 15.5% of participants lost to follow-up and the rest of the cohort in terms of baseline characteristics. The BRS total score at time point T0 had a mean score of 3.0 (SD = 1.0). At T0, 48.9% of participants scored within the "Low resilience" range (0–2.99), 40.5% scored within the "Normal resilience" range (3–4.30), and 10.5% scored within the "High resilience" range (4.31–5). At time point T1, 143 values were missing, and the mean BRS

**Table 1. Sociodemographic and academic characteristics, living conditions of the sample (N = 920).**

| | Full sample N = 920 | % / mean (SD) |
|---|---|---|
| *Sociodemographic characteristic* | | |
| Age (missing = 3) | 917 | 21.5 (3.9) |
| Gender (missing = 8) | | |
| Male | 280 | 30.4 |
| Female | 632 | 68.7 |
| Financial aid program (missing = 6) | | |
| None | 514 | 55.9 |
| Scholarship | 400 | 43.5 |
| Exchange student (yes) | 18 | 2.0 |
| Student part-time job (missing = 6) | | |
| Not applicable | 671 | 72.9 |
| Activity interrupted during lockdown | 64 | 7.0 |
| Activity increased during lockdown | 65 | 7.1 |
| No change in activity during lockdown | 114 | 12.4 |
| *Academic characteristics* | | |
| Years since starting university (missing = 4) | 916 | 3.0 (3.4) |
| Academic program (missing = 1) | | |
| Arts, humanities, languages | 114 | 12.4 |
| Law, economics, management | 126 | 13.7 |
| Social and human sciences | 176 | 19.1 |
| Sports, medical sciences, science and technology | 401 | 43.6 |
| Other | 102 | 11.1 |
| Degree pursued (missing = 2) | | |
| Undergraduate | 516 | 56.1 |
| Graduate | 272 | 29.6 |
| Other | 130 | 14.1 |
| *Living conditions* | | |
| Living arrangements | | |
| Alone | 313 | 34.0 |
| With partner | 118 | 12.8 |
| Flatsharing | 100 | 10.9 |
| With parents | 372 | 40.4 |
| Other | 17 | 1.8 |
| Someone at home had COVID-19 (missing = 2) | | |
| No | 640 | 69.6 |
| Confirmed and hospitalized cases | 14 | 1.5 |
| Confirmed and non-hospitalized cases | 164 | 17.8 |
| Suspected cases | 100 | 10.9 |
| Relative or acquaintance had COVID-19 (missing = 1) | | |
| No | 240 | 26.1 |
| Confirmed and hospitalized cases | 131 | 14.2 |
| Confirmed and non-hospitalized cases | 485 | 52.7 |
| Suspected cases | 63 | 6.8 |

**Table 2. BRS scores and resilience levels at inclusion and at first following time of measurement.**

| | Full sample N = 920 | % / mean (SD) |
|---|---|---|
| BRS total score at T0 | 920 | 3.0 (1.0) |
| Low resilience (0–2.99) | 450 | 48.9 |
| Normal resilience (3–4.30) | 373 | 40.5 |
| High resilience (4.31–5) | 97 | 10.5 |
| BRS total score at T1 (missing = 143) | 777 | 3.1 (1.0) |
| Low resilience (0–2.99) | 344 | 37.4 |
| Normal resilience (3–4.30) | 332 | 36.1 |
| High resilience (4.31–5) | 101 | 11.0 |

total score was slightly higher at 3.1 (SD = 1.0). At T1, the distribution of scores was similar to T0, with 37.4% of participants falling into the "Low resilience" category, 36.1% in the "Normal resilience" category, and 11.0% in the "High resilience" category. These findings suggest a stable distribution of resilience scores over time, with some slight changes in the proportion of participants in each resilience category.

## CFA

The goodness-of-fit indices were above the threshold of acceptability, with Satorra-Bentler-scaled-$\chi2$ = 37.86, df = 9, $\chi2$/df = 4.2, p < 0.001, and RMSEA (90% CI) = 0.059 [0.042; 0.077], SRMR = 0.022, and CFI = 0.989. The internal consistency was good, with the Cronbach's alpha at $\alpha$ = 0.89. These findings confirm previous analyses conducted in a validation study on a different sample of the same student population a year prior to the inclusions in this study [20].

## PCM

A PCM analysis of the BRS was conducted at T0 to evaluate its psychometric properties. The item difficulty estimates revealed that the scale effectively captures a range of resilience levels, with items such as BRS #1 and BRS #3 showing a broad distribution of difficulty across response categories. All items demonstrated good fit statistics, with Outfit and Infit values falling within the recommended ranges. The latent trait variance was estimated at 2.60, and the scale showed strong reliability with a Person Separation Index (PSI) of 0.90, suggesting that it effectively differentiates between individuals with varying levels of resilience. These findings support the BRS as a reliable and valid tool for measuring resilience, although some items may need refinement to improve fit for extreme response categories.

## Response shift

The longitudinal measurement invariance of the BRS was assessed using the ROSALI algorithm to determine whether response shift effects (e.g., recalibration) occurred between T0 and T1. Two models were evaluated: Model 1, which allowed for item difficulties to vary across time (non-invariance model), and Model 2, which constrained item difficulties to be equal across time (measurement invariance model).

*Step 1*

In the non-invariance model (Model 1), item difficulties were allowed to vary across the two time points. The estimates for item difficulties at T0 and T1 were similar but showed slight variations. For example, the difficulty for item BRS #1 at T0 was −3.16 (s.e. = 0.19), and at T1, it was −3.25 (s.e. = 0.22). These small differences in item difficulty across time were observed across all six BRS items. The estimates for T0 and T1 were reported for each response category of all items in Table 3.

**Table 3. Item difficulty estimates (T0 vs. T1) – Model 1 (Non-invariance).**

| Item | T0<br>Difficulty Estimate (s.e.) | T1<br>Difficulty Estimate (s.e.) |
|---|---|---|
| BRS #1 | | |
| Threshold 1 | −3.16 (0.19) | −3.25 (0.22) |
| Threshold 2 | −0.50 (0.13) | −0.68 (0.15) |
| Threshold 3 | −0.72 (0.12) | −1.15 (0.13) |
| Threshold 4 | 1.58 (0.12) | 1.80 (0.13) |
| BRS #2 | | |
| Threshold 1 | −1.44 (0.12) | −1.87 (0.14) |
| Threshold 2 | 0.70 (0.12) | 0.41 (0.13) |
| Threshold 3 | 0.37 (0.13) | −0.02 (0.13) |
| Threshold 4 | 3.17 (0.19) | 2.97 (0.18) |
| BRS #3 | | |
| Threshold 1 | −2.73 (0.16) | −3.19 (0.20) |
| Threshold 2 | 0.07 (0.12) | −0.14 (0.13) |
| Threshold 3 | −0.39 (0.12) | −0.68 (0.13) |
| Threshold 4 | 1.97 (0.13) | 2.08 (0.14) |
| BRS #4 | | |
| Threshold 1 | −2.03 (0.13) | −2.40 (0.16) |
| Threshold 2 | 0.21 (0.12) | 0.06 (0.13) |
| Threshold 3 | −0.04 (0.12) | −0.37 (0.13) |
| Threshold 4 | 2.13 (0.14) | 1.91 (0.14) |
| BRS #5 | | |
| Threshold 1 | −2.11 (0.13) | −2.30 (0.15) |
| Threshold 2 | 0.28 (0.11) | −0.02 (0.12) |
| Threshold 3 | 0.41 (0.12) | 0.33 (0.13) |
| Threshold 4 | 2.71 (0.17) | 2.57 (0.17) |
| BRS #6 | | |
| Threshold 1 | −2.09 (0.14) | −2.24 (0.16) |
| Threshold 2 | −0.04 (0.12) | −0.12 (0.13) |
| Threshold 3 | −0.10 (0.12) | −0.15 (0.13) |
| Threshold 4 | 2.53 (0.15) | 2.06 (0.15) |

The latent trait distribution estimates across the two time points showed that the variance at T0 was 2.32 (s.e. = 0.17), and at T1, it was 2.33 (s.e. = 0.18), with a covariance of 2.11 (s.e. = 0.13). These results indicate high stability in the resilience trait over time. The latent trait mean was constrained to 0 at T0, and no significant time effect was observed in the non-invariance model (time effect = 0).

*Step 2*

In the measurement invariance model (Model 2), where item difficulties were constrained to be equal across time, the item difficulty estimates were identical for T0 and T1. For instance, the difficulty for BRS #1 was −3.11 (s.e. = 0.15) at both time points, and similar stability was observed across all items (Table 4).

The latent trait variance was 2.36 (s.e. = 0.16) at T0 and 2.26 (s.e. = 0.15) at T1. The covariance between T0 and T1 was 2.10 (s.e. = 0.13). The time effect, representing the change in the latent trait mean across time, was estimated at 0.22

**Table 4. Item difficulty estimates (T0 vs. T1) – Model 2 (Measurement invariance).**

| Item | T0<br>Difficulty Estimate (s.e.) | T1<br>Difficulty Estimate (s.e.) |
|---|---|---|
| BRS #1 | | |
| Threshold 1 | −3.11 (0.15) | −3.11 (0.15) |
| Threshold 2 | −0.48 (0.10) | −0.48 (0.10) |
| Threshold 3 | −0.82 (0.09) | −0.82 (0.09) |
| Threshold 4 | 1.79 (0.10) | 1.79 (0.10) |
| BRS #2 | | |
| Threshold 1 | −1.53 (0.10) | −1.53 (0.10) |
| Threshold 2 | 0.67 (0.09) | 0.67 (0.09) |
| Threshold 3 | 0.28 (0.10) | 0.28 (0.10) |
| Threshold 4 | 3.17 (0.14) | 3.17 (0.14) |
| BRS #3 | | |
| Threshold 1 | −2.83 (0.13) | −2.83 (0.13) |
| Threshold 2 | 0.07 (0.10) | 0.07 (0.10) |
| Threshold 3 | −0.42 (0.10) | −0.42 (0.10) |
| Threshold 4 | 2.12 (0.11) | 2.12 (0.11) |
| BRS #4 | | |
| Threshold 1 | −2.09 (0.11) | −2.09 (0.11) |
| Threshold 2 | 0.24 (0.10) | 0.24 (0.10) |
| Threshold 3 | −0.09 (0.10) | −0.09 (0.10) |
| Threshold 4 | 2.12 (0.11) | 2.12 (0.11) |
| BRS #5 | | |
| Threshold 1 | −2.10 (0.11) | −2.10 (0.11) |
| Threshold 2 | 0.24 (0.09) | 0.24 (0.09) |
| Threshold 3 | 0.48 (0.10) | 0.48 (0.10) |
| Threshold 4 | 2.74 (0.13) | 2.74 (0.13) |
| BRS #6 | | |
| Threshold 1 | −2.06 (0.11) | −2.06 (0.11) |
| Threshold 2 | 0.02 (0.09) | 0.02 (0.09) |
| Threshold 3 | −0.02 (0.09) | −0.02 (0.09) |
| Threshold 4 | 2.40 (0.11) | 2.40 (0.11) |

(s.e. = 0.04) and was statistically significant (p < 0.0001). This suggests a small but significant increase in resilience over the study period.

A likelihood-ratio test comparing the two models (Model 1 vs. Model 2) yielded a chi-square statistic of 25.05 (df = 23, p = 0.3479), indicating that the difference between the models was not statistically significant. This result supports the conclusion that the assumption of measurement invariance holds, and there is no significant recalibration or response shift across time points.

## Discussion

The present study examined the longitudinal measurement invariance of the French version of the Brief Resilience Scale across two time points, using the ROSALI algorithm to detect potential response shift effects (e.g., recalibration) in university students during the COVID-19 pandemic. The results suggest that the BRS-F demonstrated measurement invariance

across the two time points, indicating that changes in resilience observed over time were not attributable to response shifts in the way participants evaluated their resilience.

Our analysis revealed no significant recalibration or response shift effects in the BRS-F items, as indicated by the likelihood-ratio test and the similarity in item difficulty estimates between T0 and T1 under the measurement invariance model (Model 2). In other words, the perceived difficulty of items remained consistent across the two time points, supporting the conclusion that the changes in resilience reported by participants were genuine, rather than a product of changes in how they interpreted the scale's items. These findings are consistent with the assumption that resilience, as measured by the BRS-F, is a relatively stable construct over short periods, even in the face of significant stressors such as the COVID-19 pandemic.

Previous psychometric research on resilience measurement has consistently shown that, despite conceptualizing resilience as a dynamic and context-dependent process, commonly used resilience scales tend to exhibit robust measurement properties, including stable factor structures, good internal consistency, and evidence of measurement invariance across groups [16–21]. For example, validation studies of instruments such as the Brief Resilience Scale have reported satisfactory reliability and structural stability, supporting their use for comparing resilience levels across individuals and contexts [18–20]. Importantly, this psychometric robustness does not imply that longitudinal measurement invariance has been firmly established. Rather, it suggests that while individuals' levels of resilience may fluctuate in response to changing circumstances, the meaning of the items and the underlying measurement model may remain sufficiently consistent to allow valid longitudinal comparisons. This assumption requires explicit testing, and the present findings directly address this gap by demonstrating that, over a short time frame and during a period of significant contextual stress, the French version of the BRS maintained longitudinal measurement invariance without evidence of response shift.

It is important to note that resilience measures differ in the extent to which they reflect the dynamic nature of resilience. Some widely used instruments primarily assess relatively stable personal characteristics or protective resources (e.g., persistence, self-efficacy, sense of purpose), which may be less sensitive to short-term contextual changes [16,17]. In contrast, the BRS was explicitly designed to capture a functional recovery process, namely, the perceived ability to bounce back from stress, rather than stable traits or enduring resources [18]. This focus on recovery allows the BRS to reflect changes in resilience levels as individuals adapt to evolving stressors, while relying on items whose meaning remains sufficiently stable across time. In this sense, the dynamism of resilience is expressed through changes in the latent trait, whereas measurement stability is preserved through a consistent interpretation of items content. The absence of response shift observed in the present study supports this distinction, suggesting that the BRS is sensitive to short-term adaptive changes without being confounded by shifts in respondents' internal standards or conceptualization of resilience.

The analysis revealed a statistically significant time effect ($p < 0.0001$), with a slight but meaningful increase in the latent trait of resilience over the study period. This suggests that, despite the stability of the scale's measurement properties, university students did experience a genuine increase in resilience during the pandemic. The time effect estimate of 0.22 indicates that participants' resilience levels increased slightly between the two time points, which could reflect adaptations to the ongoing crisis or a coping response to the pandemic's challenges.

This increase in resilience over time could be interpreted as part of a broader pattern of psychological adaptation in response to prolonged stress. Previous research has shown that individuals can adapt to chronic stressors and become more resilient as they develop new coping strategies [3,9]. The COVID-19 pandemic, with its unprecedented challenges, may have provided students with opportunities to build or reinforce their resilience, either through personal growth or by relying on new social and academic resources. However, it is important to note that while the change in resilience was statistically significant, it was relatively small, and further research would be needed to explore the long-term trajectory of resilience in this population.

The findings of this study have important implications for the psychometric assessment of resilience. Given the lack of evidence for response shift effects, it appears that the BRS-F can be used to reliably assess resilience in university

students across time, even during periods of significant disruption like the COVID-19 pandemic. However, the significant increase in resilience observed warrants further investigation into the factors contributing to this change. Specifically, future studies could explore the role of social support, coping strategies, and academic engagement in fostering resilience over time.

It is also worth noting that while this study focused on the Brief Resilience Scale, the findings may be relevant for other resilience measurement tools. The methodology used in this study, including the ROSALI algorithm, could be applied to other scales to assess their longitudinal measurement invariance, helping to ensure the accuracy and validity of resilience assessments across time.

## Strengths, limitations, and future research

This study has several strengths that enhance its robustness and validity. First, the use of the ROSALI-RMT algorithm provides a rigorous approach to examining longitudinal measurement invariance, with lower rates of false response shift detection compared to other methods [34]. This ensures that the observed changes in resilience are genuine, rather than artifact-based, increasing the study's credibility. Additionally, the study was conducted during the COVID-19 pandemic, offering valuable insights into how resilience evolves in response to a global crisis, particularly among university students – a group facing significant disruption. The large sample size further strengthens the study's reliability, reducing the risk of sampling bias and improving the generalizability of the findings. Moreover, using the well-validated BRS-F, combined with advanced methods like ROSALI-RMT, reinforces the study's conclusions about the psychometric properties of the BRS in assessing resilience during a crisis.

However, there are also limitations. The ROSALI-RMT algorithm is limited to detecting recalibration, and does not capture other types of response shift, such as reprioritization or reconceptualization, which may have occurred unnoticed. While the large sample size is a strength, the study's focus on university students from a specific region limits its generalizability to other populations. Additionally, although 15.5% of values were missing at the second measurement time, the use of IRT modeling, and particularly the ROSALI-RMT algorithm, ensures robustness against this level of missingness, even under non-random conditions. Nonetheless, the influence of missing data cannot be entirely ruled out and should be acknowledged when interpreting the results.

The relatively short time frame between measurements, which was chosen to limit attrition and ensure sufficient longitudinal data, also suggests that observed changes in resilience might reflect short-term adaptations, rather than long-term shifts in resilience. Lastly, while the ROSALI algorithm is a powerful tool for assessing measurement invariance, it relies on the assumption that the underlying construct of resilience is stable across time.

Future research could expand upon the findings of this study by exploring resilience in a broader range of populations, including individuals from different educational backgrounds, cultural contexts, and geographical locations. This would help to better understand how resilience evolves across diverse groups and whether the patterns observed in this study are generalizable. Additionally, the relatively short time frame between measurements in this study calls for future investigations with longer follow-up periods. Such studies would provide insights into whether resilience gains observed during a crisis are sustained or whether they diminish over time. It would also be valuable to explore other forms of response shifts, such as reprioritization and reconceptualization, which were not captured in this study. Incorporating methods capable of detecting these shifts could offer a more comprehensive understanding of how individuals' perceptions of resilience change in response to long-term stressors. Lastly, future studies could examine whether resilience evolves differently for different individuals, considering personal characteristics or coping strategies, and whether the BRS remains sensitive enough to capture these individual variations. Expanding the scope of resilience research could deepen our understanding of how people adapt to both short-term and prolonged challenges.

## Conclusions

This study provides strong evidence that the Brief Resilience Scale (BRS) maintains measurement invariance over time, supporting its reliability for tracking resilience even during major disruptions like the COVID-19 pandemic. The absence of response shift effects confirms that observed changes reflect true psychological adaptation, not measurement bias.

The modest increase in resilience highlights students' adaptive capacity, while also suggesting variability in how individuals experience and respond to stress. However, this finding should be interpreted with caution and verified through further longitudinal studies, as it remains essential to determine whether such changes are consistent, context-dependent, or influenced by individual factors over time. Identifying distinct resilience profiles could guide more tailored mental health support.

Practically, the BRS can serve as a reliable tool for monitoring resilience in university settings. Institutions could use these insights to implement targeted interventions, such as resilience workshops, peer mentoring, or academic support—especially during crises. These findings support both the valid use of resilience measures over time and the development of informed, student-centered well-being strategies.

## Acknowledgments

The authors extend their sincere gratitude to all study participants for their time, engagement, and valuable contributions. We also thank Dr. Myriam Blanchin for her invaluable guidance and technical support with the PRO-online platform.

Consent to participate: Informed consent was obtained from all individual participants included in the study.

## Author contributions

**Conceptualization:** Anja Todorović, Cédric Baumann, Christine Rotonda, Sarah Lapcevic, Stéphanie Bourion-Bédès.

**Data curation:** Anja Todorović, Elise Eby.

**Formal analysis:** Anja Todorović.

**Investigation:** Anja Todorović.

**Methodology:** Anja Todorović, Cédric Baumann, Christine Rotonda, Sarah Lapcevic, Stéphanie Bourion-Bédès.

**Supervision:** Cédric Baumann, Stéphanie Bourion-Bédès.

**Validation:** Anja Todorović, Cédric Baumann, Christine Rotonda, Elise Eby, Stéphanie Bourion-Bédès.

**Writing – original draft:** Anja Todorović.

**Writing – review & editing:** Anja Todorović, Cédric Baumann, Christine Rotonda, Elise Eby, Sarah Lapcevic, Stéphanie Bourion-Bédès.

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
