## [Decision Letter · Decision Letter 0]

11 Nov 2025

Dear Dr. Todorović,

Thank you for submitting your manuscript to PLOS ONE. After careful consideration, we feel that it has merit but does not fully meet PLOS ONE’s publication criteria as it currently stands. Therefore, we invite you to submit a revised version of the manuscript that addresses the points raised during the review process.

We look forward to receiving your revised manuscript.

Kind regards,

Charles Martin-Krumm, Ph.D.

Academic Editor

PLOS ONE

“The authors extend their sincere gratitude to all study participants for their time, engagement, and valuable contributions. We also thank Dr. Myriam Blanchin for her invaluable guidance and technical support with the PRO-online platform. This research is a part of a PhD project funded by the French Network of Doctoral Programmes in Public Health (RDSP), coordinated by EHESP French School of Public Health.”

Reviewers' comments:

Reviewer's Responses to Questions

**Comments to the Author**

1. Is the manuscript technically sound, and do the data support the conclusions?

Reviewer #1: Yes

Reviewer #2: Yes

2. Has the statistical analysis been performed appropriately and rigorously?

Reviewer #1: I Don't Know

Reviewer #2: Yes

3. Have the authors made all data underlying the findings in their manuscript fully available?

Reviewer #1: Yes

Reviewer #2: Yes

4. Is the manuscript presented in an intelligible fashion and written in standard English?

Reviewer #1: Yes

Reviewer #2: Yes

Reviewer #1: Minor edits:

Lines 52-53: “University students, who are often in a period of transition and identity formation.”

[Add reference/source]

Lines 58-60: “Studies showed that individuals with higher levels of resilience were better equipped to cope with pandemic-related stressors and reported fewer mental health symptoms (7).”

[Only one reference/source, yet you mention “studies”, which I guess are multiple? Consider adding more sources, and even potentially developing slightly that insight.]

Lines 108-112: “The survey included questions on students' sociodemographic characteristics, their living and learning conditions during the lockdown, the perceived health threat of COVID-19 in their immediate environment, the impact of lockdown and related measures on their living and learning conditions, as well as self-administered questionnaires assessing their health status.”

[Perhaps not necessary here in the main body, but those included questions could eventually be made accessible to the reader via a link or a supplementary material file. Or perhaps you could add something like "available via corresponding author, on demand"]

Moderate edits:

Lines 41-42: “Psychological resilience is the capacity to adapt and recover from adversity, enabling individuals to maintain their psychological well-being in the face of stress and challenges.”

[Add reference/source that clearly states that. Also, it is not a wrong definition per se, but the “maintain psychological well-being” aspect is debatable… If you “recovered” from something, can it mean that you “maintained” well-being during the whole process without it being impacted at all at any point? I would advise going slightly deeper about resilience (either in the background through your readings for a more comprehensive understanding, or here in the introduction section by adding a few sentences)]

Lines 60-61: “This underscores the importance of resilience in helping individuals maintain stability during times of crisis.”

[Lack of explanation here. Why precisely resilience is of help to maintain stability? Which resilience factors help, or what aspect/skill/capacity helps facing such adversity?]

Lines 65-66: “This has led to the development of various resilience measurement tools, such as the Brief Resilience Scale (BRS) (10). It is essential to assess whether these tools maintain consistent validity over time”

[Why is the BRS the only scale mentioned? How does it represent the shift toward a modern approach to resilience (as a dynamic and evolving process)? What are the other scales available, and what are their differences? Have you considered exploring some of the advantages and disadvantages between scales in order to justify the choice of the BRS? Are this choice and these details already addressed in another publication related to the PIMS2-SPE study? If so, it would be appropriate to reference it here. Here is a link leading to literature about several measurement tools]

https://scholar.google.com/scholar?hl=fr&as_sdt=0%2C5&q=measuring+and+assessing+resilience&btnG=

Lines 94-95: “The measurement times used for this study were T0 and T1.”

[To me, more details on why those precise measurement times were chosen (T0 and T1) would seem important to display here.]

Lines 375-376: “The relatively short time frame between measurements also suggests that observed changes in resilience might reflect short-term adaptations, rather than long-term shifts in resilience.”

[Good point here. As mentioned above, it would be interesting to add explanation on why you chose such a short time frame between the two measurements.]

Major edits:

Lines 75-76: “no previous study has assessed the longitudinal measurement invariance of resilience scales, particularly in relation to the potential effects of response shift (12,13).”

[This to me seems a little hasty. More precision is needed here. Perhaps effects of response shift have not been explored previously, yet is it true for the invariance of resilience scales? Perhaps invariance in terms of effects of response shift has not been studied, but then a distinction needs to be worded here. A quick search with the right keywords made me doubt, or at least question, such lack of previous body of work you put forward here. Here is the link to literature exploring measurement invariance of resilience scales]

https://scholar.google.com/scholar?oi=gsb95&q=invariance%20resilience%20scale&lookup=0&hl=fr

Lines 80-81: “We hypothesized that the BRS, when used across multiple time points, would maintain its psychometric properties, enabling valid comparisons of resilience over time.”

[Why such a hypothesis? On what elements and/or theoretical and empirical insights do you base your hypothesis on?]

Recap for the whole introduction section:

[I have not checked the author guidelines; therefore, I don't know if you would have room and could be allowed for a lengthier introduction... Yet if so, I would globally consider adding more depth to this whole introduction section...

- What are the different tools/scales for measuring resilience?

- Or at least, what are some of the more robust and well-established ones?

- What are their advantages/disadvantages, or what are some of their differences?

- What are the ones/the one that align the best with modern approaches considering resilience as a dynamic process?

- Why choosing the BRS?]

Lines 323-324: “This aligns with earlier studies on resilience measurement, which suggest that while resilience is a dynamic process that can evolve over time, it is also stable enough to be measured consistently across time points (10).”

[This needs to be more developed and discussed. Earlier studies on resilience measurement have not been mentioned (nor reflected upon) enough throughout the paper (introduction, and here in the discussion). I would advise to at least give some insights here on the results and implications from those earlier studies in order then to compare them to your own results and in order to offer a more comprehensive and complete discussion. Also there needs to be more clarity on the fact resilience is said to be a dynamic process while being stable enough to be measured across time points. To me your assertion here seems too hasty and needs to be developed and explored.]

Recap for the whole discussion section:

[This discussion section could point to some missing elements/aspects in your introduction (elements that perhaps you'll add in the intro so that you'll be able to link them to your discussion here). More globally, it could be interesting to explore some of the following questioning:

- Do all resilience scales encapsulate the dynamic nature of resilience?

- How does the BRS position itself in this regard?

- What dimensions/factors of the BRS and of other well-known scales (in those earlier studies maybe) account for the dynamism of the resilience construct?

- At the same time, how come those dimensions/factors/items account for some sort of stability in the results across time points?]

GENERAL AND/OR ADDITIONAL COMMENT:

[I sure understand you are not writing a theoretical paper about resilience, neither that you aim at discussing conceptual boundaries of such a construct and the tools developed to measure it, so no worries here. You do not need to rethink the whole approach behind your study. My point is that, to me, a more comprehensive reflection and a deeper discussion need to be displayed in your paper about these aspects (results from earlier studies; comparison with your results; dynamism of resilience while scales lead to stable results; resilience dimensions targeted within those measuring tools; and how all of these insights are to be interpreted altogether...)]

Reviewer #2: The ethical standards of the study were respected (according to the principles of the Declaration of Helsinki).

The study framework, the composition of the study population sample, and the data collection and processing were carried out rigorously and clearly explained within the article.

The bibliographic references are robust.

The results were thoroughly analyzed, and the data interpretation was performed accurately and cautiously.

In conclusion, this article demonstrates sound scientific quality and is suitable for publication as is.

**Do you want your identity to be public for this peer review?** For information about this choice, including consent withdrawal, please see our For information about this choice, including consent withdrawal, please see our Privacy Policy .

Reviewer #1: No

Reviewer #2: No

---

## [Author Response · Author response to Decision Letter 1]

18 Jan 2026

Dear Pr. Martin-Krumm,

We thank you and the reviewers for the careful evaluation of our manuscript and for the constructive feedback provided. We are pleased to submit a revised version of our manuscript that addresses all points raised by the academic editor and the reviewers.

In response to Reviewer #1’s comments, we substantially strengthened the Introduction and Discussion sections. Specifically, we clarified the conceptualization of psychological resilience, expanded the discussion of resilience measurement tools, and more explicitly justified the choice of the Brief Resilience Scale in light of contemporary, process-oriented approaches to resilience. We also refined our claims regarding longitudinal measurement invariance, clearly distinguishing between previously established psychometric robustness and the relative lack of response shift-informed longitudinal evidence. In the Discussion, we developed a more integrated reflection on how resilience can be understood as a dynamic construct while still yielding stable measurement properties, and we explicitly positioned our findings in relation to earlier psychometric studies.

Reviewer #2’s positive evaluation is gratefully acknowledged.

In accordance with the Academic Editor’s additional requirements:

• The manuscript has been revised to comply with PLOS ONE formatting and style guidelines.

• The ethics statement has been consolidated and appears exclusively in the Methods section. We now clearly specify that informed consent was obtained electronically prior to participation, in line with institutional and GDPR requirements, and the same text has been entered in the online submission form.

• All funding-related information has been removed from the Acknowledgments section. We kindly request that the Funding Statement be updated to reflect the following:

Funding Statement (revised):

This research is a part of a PhD project funded by the French Network of Doctoral Programmes in Public Health (RDSP), coordinated by EHESP French School of Public Health. The funders had no role in study design, data collection and analysis, decision to publish, or preparation of the manuscript.

All changes made to the manuscript are highlighted in the tracked-changes version, and a detailed, point-by-point rebuttal is provided in the accompanying “Response to Reviewers” document.

We believe that the revised manuscript now fully addresses the reviewers’ and editor’s concerns while remaining faithful to the scope and aims of the study. We sincerely thank you for the opportunity to revise our work and hope that the manuscript is now suitable for publication in PLOS ONE.

Kind regards,

Dr. Anja Todorović

(on behalf of all authors)

---

## [Decision Letter · Decision Letter 1]

9 Mar 2026

Resilience Across Time: Longitudinal Measurement Invariance of the Brief Resilience Scale in French University Students.

PONE-D-25-29756R1

Dear Dr. Todorović,

We’re pleased to inform you that your manuscript has been judged scientifically suitable for publication and will be formally accepted for publication once it meets all outstanding technical requirements.

Kind regards,

Frantisek Sudzina

Academic Editor

PLOS One

Additional Editor Comments (optional):

Reviewers' comments:

Reviewer's Responses to Questions

**Comments to the Author**

Reviewer #1: All comments have been addressed

Reviewer #2: All comments have been addressed

2. Is the manuscript technically sound, and do the data support the conclusions?

Reviewer #1: Yes

Reviewer #2: Yes

3. Has the statistical analysis been performed appropriately and rigorously?

Reviewer #1: Yes

Reviewer #2: Yes

4. Have the authors made all data underlying the findings in their manuscript fully available?

Reviewer #1: Yes

Reviewer #2: Yes

5. Is the manuscript presented in an intelligible fashion and written in standard English?

Reviewer #1: Yes

Reviewer #2: Yes

Reviewer #1: I would like to congratulate the authors on their work and on their efforts to address all the points and needed edits raised earlier in the review process. They added clarity, depth and made the manuscript more robust overall (in particular the introduction and discussion sections, as their method and data analysis were already sound). Thank you also for taking the time to clearly reply to each of my comments by explaining how you addressed each of them. I now recommend this work for publication.

Best regards,

Reviewer #1

Reviewer #2: The authors of the article responded with great clarity and attention to the requests for clarification they received after the initial review. The article, thus revised, is, in my opinion, entirely suitable for publication.

**Do you want your identity to be public for this peer review?** For information about this choice, including consent withdrawal, please see our For information about this choice, including consent withdrawal, please see our Privacy Policy .

Reviewer #1: **Yes:** Clément MétaisClément Métais

Reviewer #2: No

---

## [Editor Report · Acceptance letter]

PONE-D-25-29756R1

PLOS One

Dear Dr. Todorović,

I'm pleased to inform you that your manuscript has been deemed suitable for publication in PLOS One. Congratulations! Your manuscript is now being handed over to our production team.

Kind regards,

on behalf of

Dr. Frantisek Sudzina

Academic Editor

PLOS One